# Spatial–Temporal Features and Correlation Studies of County Rural Hollowing in Sichuan

**DOI:** 10.3390/ijerph19159075

**Published:** 2022-07-26

**Authors:** Shili Guo, Qiuyan Chen, Yao He, Dingde Xu

**Affiliations:** 1School of Economics, Southwestern University of Finance and Economics, Chengdu 611130, China; guoshili@swufe.edu.cn; 2West Center for Economic Research, Southwestern University of Finance and Economics, Chengdu 611130, China; yaohe2021@smail.swufe.edu.cn; 3College of Management, Sichuan Agricultural University, Chengdu 611134, China

**Keywords:** rural hollowing, Sichuan Province, temporal and spatial evolution

## Abstract

The research on rural hollowing is necessary for policy making related to rectifying hollow villages in the Rural Revitalization Strategy. In this context, Sichuan was chosen as a typical region to explore the spatial–temporal features and relevant factors of rural hollowing. The results show that (1) from 2010 to 2018, rural hollowing shows a downward trend. The average of comprehensive hollowing dropped by 0.028, and 119 counties have a decrease in the degree of comprehensive hollowing. (2) During the same study period, the regional characteristics of the rural hollowing degree were significant. High and higher degrees of comprehensive hollowing have an obvious decrease in number and have a trend of agglomeration in Central Sichuan. High and higher degrees of land hollowing have an obvious decrease in number, and they were mainly distributed in Northwest Sichuan in 2018. The degree of population hollowing gradually increases from west to east, and high and higher degrees of population hollowing have a significant increase in number, mainly concentrated in East Sichuan. The degree of economic hollowing has obvious spatial characteristics from west to east, and high and higher degrees of economic hollowing have an increase in number, mainly concentrated in Chengdu Plain. (3) During the same study period, the spatial distribution characteristics of rural hollowing degree at the county level in Sichuan Province were obvious. The high-value clustering is mainly concentrated in Chengdu Plain Economic Zone, and the range of clustering is decreasing. Low-value clustering was mainly distributed in Northeast Sichuan Economic Zone and Northwest Sichuan Economic Zone. (4) Rural hollowing in Sichuan Province is negatively correlated with average elevation and per capita arable land area; rural hollowing is positively correlated with urbanization rate and industrial output value.

## 1. Introduction

Urban and rural sustainable development is one of the goals of global sustainable development [1]. However, when China’s urbanization and industrialization are rapidly advancing, the phenomenon of rural hollowing appears. Rural hollowing, which is described as “outward expansion with inside hollowing”, is a phenomenon that is harmful to sustainable rural development in the process of urbanization and industrialization. It encompasses scenarios such as one family owning more houses, building new houses without dismantling the old one, deserting new houses [2]. To put it simply, in the process of urbanization and industrialization, the area of rural settlements has not been accompanied by a decrease in the rural population [3]; on the contrary, the per capita residential land in rural areas continues to increase [4]. Rural hollowing wastes precious land, causing the loss of rural talents, funds and other resources, and destroying the rural living environment [5]. The renovation of hollow villages is an indispensable stage to realizing the strategy of rural revitalization, and it is also the premise of sustainable rural development. Additionally, rural hollowing assessment is the premise and foundation of hollow village renovation, so rural hollowing assessment is very important.

In recent years, rural hollowing has attracted attention from demography, geography, sociology and other disciplines. The study of rural hollowing generally focuses on the macro level and looks at the problem from the macro perspective. The study of rural hollowing mainly focuses on the definition, characteristics, influencing factors, response and regularity of rural hollowing [6], and rural settlements [7]. Meanwhile, studies also pay attention to rural hollowing degrees from population, land and economy [7,8,9,10].

The push–pull theory is an important theory to explain rural hollowing [2]. The higher income, non-agricultural employment, etc., in cities are regarded as centrifugal forces, and the local complex, rural social relations, etc., are regarded as centripetal forces. When the centrifugal force is greater than the centripetal force, the phenomenon of rural to urban migration will occur. After China’s reform and opening up, the household registration system began to weaken, industrialization and urbanization proceeded rapidly, and farmers were given the opportunity to leave agriculture and the countryside [11]. At this time, the strong centrifugal force formed by higher income and non-agricultural employment opportunities in the cities is far greater than the centripetal force [2]. Therefore, China’s rural population has begun to migrate to cities on a large scale. The loss of permanent population in rural areas is serious, and rural production and economy are seriously declining [12]. After farmers go out to work, their income also increases, their ability to improve their living conditions is significantly improved, and the area of rural homesteads continues to expand [5]. Moreover, because of the collective ownership system of rural land, once farmers settle down in cities, it is difficult to trade rural housing, resulting in homesteads abandonment [5].

Just like the push–pull theory, the current theoretical mechanism of rural hollowing is more concentrated in three aspects: economy, system and policy. Some scholars have also tried to incorporate resources and environment into the formation mechanism of rural hollowing, but they have emphasized the impact of natural disasters, cultivated land resources, and surrounding open space on rural hollowing [5,11]. In this paper, the elevation is included in the formation mechanism of rural hollowing, trying to improve the rural hollowing mechanism.

The evaluation index system of rural hollowing degree in academics is mostly measured from land, population and economic system. In terms of the land subsystem, indicators such as the extensive utilization of homesteads, the transformation degree of homestead utilization, the relative diffusion of villages and residential land area are selected more frequently [2,10,13,14]. The hollowing of the rural population is the emphasis of the current academic research, and its main measurement indicators include the proportion of the rural resident population, the effective transfer degree of the village population, the intensive degree of the rural population, and the centrality of the rural population, etc. [8,14,15,16,17]. In the economic subsystem, scholars often use indicators such as economic development level, farmers’ income level, and regional economic structure [8,18]. Some scholars use electricity consumption per household to measure hollowing out [18,19]. The details of the indicators are shown in Table 1.

When evaluating the degree of rural hollowing out, most of them adopt the entropy method or the analytic hierarchy process, combined with the entropy weight method to give the index weight [8,16,18,20]. Some scholars also use the BP neural network model to determine the weight [9]. Longhualou et al. took the lead in measuring the degree of rural hollowing by using four indicators of homestead vacancy rate, abandonment rate, percentage of permanent residents, and village population density [17]. These indicators are simple, clear and representative, but the data on the vacancy rate and abandonment rate of homesteads need to be acquired through field research, which limits the expansion of the research area and the study of long-term trends.

Yang Ren et al. used seven indicators to measure the degree of rural hollowing from the three subsystems of land, population and economy, including the transformation degree of homestead utilization, the relative diffusion of villages, the extensive utilization of homestead, the centrality of rural population, the intensive degree of rural population, effective transfer degree of the village population, the level of economic development, the regional economic structure and the income level of farmers [9]. This evaluation index system has been widely recognized and solved the difficulty of obtaining data excluding extensive utilization of homestead, but it has also been questioned, mainly focusing on the two indicators of the centrality of the rural population and the degree of effective transfer of village population [21].

**Table 1 ijerph-19-09075-t001:** Representative indicators of rural hollowing.

Subsystem	Indicators	Reference
Land	Homestead vacancy rate, homestead abandonment rate	[17]
Land	residential land area, residential area	[2]
Land	The transformation degree of homestead utilization, the relative diffusion of villages, the extensive utilization of homestead	[9]
Land	Vacant residential land area, abandoned residential land area	[13]
Land	Relative size of homestead per capita, homestead area ratio, relative rate of change in village size	[16]
Land	The total amount of homestead, number of abandoned and vacant homesteads	[22]
Land	Extensive utilization of homestead, hollowing trend	[20]
Land	Per capita residential land area, the transformation degree of homestead utilization, relative size per capita, road density	[19]
Land	Land reclamation rate, intensity of settlements, food production per unit area, marginalization rate of cultivated land, fixed asset investment, power of agricultural machinery	[8]
Land	Rural settlement area per capita	[14]
Land	Village relative spread, extensive utilization of homestead, the transformation degree of homestead utilization	[23]
Population	The proportion of rural resident population, village population density	[17]
Population	agricultural labor force structure	[15]
Population	Rural population centrality, the intensive degree of rural population, effective transfer degree of village population	[9]
Population	Rural land population density, rural population size, the relative proportion of non-agricultural population	[16]
Population	Rural population centrality, Farmer’s effective transfer degree, population change rate, village population concentration	[7]
Population	Percentage of rural permanent population, share of non-agricultural population, rural population structure	[19]
Population	Urbanization rate, the proportion of rural employees, rural population settlement rate, population agglomeration	[8]
Population	Effective transfer degree of village population, rural population centrality, rural population agglomeration	[23]
Economy	The level of economic development, per capita net income of farmers	[15]
Economy	The level of economic development, regional economic structure, farmer’s income level	[13]
Economy	Regional economic structure, urban–rural income ratio	[20]
Economy	Electricity consumption per household	[19]
Economy	The level of economic development, regional economic structure, farmer’s income level	[8]

The subsequent improvement of the rural hollowing index system is based on them. Many new indicators make the rural hollowing evaluation system more reasonable, such as the relative size of per capita homestead [16], the ratio of urban and rural income [20], and the electricity consumption per household [19].

In this research, the index of electricity consumption per household is improved, hoping to improve the accuracy of the evaluation index system. Zhang Xiupeng et al. used electricity consumption per household to measure the degree of rural hollowing. They believe that the higher the per household electricity consumption of the entire county, the longer the average person spends at home, and the lower the rural hollowing [19]. This research uses rural per capita electricity consumption instead. The higher the rural per capita electricity consumption, the longer the average person spends at home in rural areas, and the lower the degree of rural hollowing. Using rural per capita electricity consumption to measure rural hollowing is significantly more in line with the connotation and improves its accuracy.

China has launched the ‘increasing vs. decreasing balance’ land-use policy in order to optimize rural land use. In 2006, Sichuan Province was included in the pilot areas for the ‘increasing vs. decreasing balance’ land-use policy. It has achieved remarkable results in revitalizing and utilizing residential land. However, with the continuous advancement of the urbanization process, the problems of multiple houses in one family and vacant houses in rural areas are more prominent due to reasons such as migrant work, settlement, and inheritance. Residential lot area is getting larger and larger, and the per capita residential land area in the province’s rural areas far exceeds the specified area [24]. Moreover, Sichuan Province is a large province of migrant workers [25]. Therefore, Sichuan Province is a typical region for the study of rural hollowing. This research can provide a scientific basis for the renovation of hollow villages in the Sichuan Province.

As mentioned above, this paper explores the formation mechanism of rural hollowing, especially adding altitude to it, and hopes to improve the forming mechanism of rural hollowing. Additionally, improved the evaluation index of rural hollowing, hoping to improve the evaluation method of rural hollowing. On this basis, urbanization, industrialization, arable land area, and average elevation were selected for correlation analysis, and the impact mechanism of rural hollowing was initially discussed empirically. The specific objectives of this study are as follows:The overall change pattern of rural hollowing in Sichuan in 2010, 2015 and 2018.The change characteristics of rural hollowing in the study area during the same study period.The correlation of resources and environment, urbanization and industrialization on rural hollowing.

## 2. Theoretical Analyses

As shown in Figure 1, farmers have been pursuing a better life, and improving their living conditions is one of their pursuits. However, due to the low efficiency of traditional agriculture, the ability to improve housing is very poor.

The huge gap between urban and rural areas and poor natural conditions makes farmers want to leave the countryside and traditional agriculture. Industrialization and urbanization have given farmers the opportunity that they can go out to work and settle in cities.

The migration of farmers to work and settle in cities leads to population loss, which directly leads to population hollowing. In addition, the loss of population leads to the reduction of professional farmers, which in turn leads to economy hollowing.

After farmers go out to work, their income increases, their ability to improve their housing is improved, the area of rural homesteads expands, and farmers go out to work for a long time, resulting in long-term idle houses. After farmers settled in cities, due to the collective ownership of rural land, it was difficult to trade rural houses, which led to the abandonment of rural houses. These behaviors have led to land hollowing.

Therefore, this paper selects four indicators of average elevation, per capita arable land, urbanization rate and industrial output value, and uses the Pearson correlation test to preliminarily explore the relevant factors of rural hollowing from three aspects: resource and environmental endowment, urbanization and industrialization.

## 3. Data and Methods

### 3.1. Study Area

Sichuan Province is located in southwest China, with an area of 486,000 square kilometers. It has 21 cities (prefectures) and 183 counties (cities, districts) under its jurisdiction. The geographical features of the east and west of the province are largely different, and the overall feature is that the west is high and the east is low (Figure 2). The west is mostly plateaus and mountains, with an average elevation of more than 4000 m; meanwhile, the east is mostly hilly, with an altitude between 1000 and 3000 m (Source: Sichuan Province Geographical Map). At the end of 2019, the GDP of Sichuan Province was RMB 4661.58 billion, and the total output value of agriculture, forestry, animal husbandry and fishery reached RMB 788.935 billion. The current permanent resident population is 83.75 million, and the registered population is 90.995 million. From 2016 to 2019, the rural registered population continued to decrease, from 61.4 million to 57.53 million, and the urban population increased from 27.852 million to 33.468 million (Source: Sichuan Statistical Yearbook in 2019). Meanwhile, the number of rural labor transfer employment in Sichuan reached 24.826 million (Source: Sichuan Statistical Yearbook in 2019). The province can be divided into five major economic zones: Chengdu Plain, northeast Sichuan, south Sichuan, Panxi and northwest Sichuan.

### 3.2. Data

After collecting and sorting out the socio-economic statistical data of 183 districts and counties in Sichuan Province, we discovered that the proportion of the non-agricultural population in Jinjiang District, Qingyang District, Jinniu District, Wuhou District and Chenghua District of Chengdu reached 100% in 2010, and the urbanization rate reached 100% in 2013. Therefore, it is of little significance to study the degree of rural hollowing, so these places are not within the scope of this paper.

The socio-economic data of Sichuan Province in this paper comes from China Statistical Yearbook (http://www.stats.gov.cn/tjsj/ndsj/, accessed on 1 January 2019) and Sichuan statistical yearbook (http://tjj.sc.gov.cn/scstjj/c105855/nj.shtml/, accessed on 1 January 2019). The total administrative area, rural residential area, construction land area, and elevation of 178 counties in Sichuan Province in 2010, 2015 and 2018 are derived from the remote sensing monitoring data of China’s land use released by the Resource and Environmental Science and Data Center of the Chinese Academy of Sciences (https://www.resdc.cn/, accessed on 1 January 2019). In this paper, the rural residential area in 2009, 2014 and 2017 is calculated by the annual average growth rate from 2010 to 2018. If the population-related indicators involved in this paper are not counted, they are also calculated by the annual average growth rate.

### 3.3. Methods

#### 3.3.1. Degree of Rural Hollowing Index

It is difficult to obtain the residential land utilization data of all villages in Sichuan Province. Therefore, taking improvements of previous research and the availability of data into account, this research used the comprehensive correlation and operability principles, selecting typical and representative indicators to construct the evaluation index system of the rural hollowing degree in Sichuan Province from the perspective of the county. According to the connotation of rural hollowing defined by Liu Yansui [2], this paper selects the following indicators to measure the degree of rural hollowing in Sichuan Province, referring to the research results of scholars. Indicators are provided in Table 2.

Transformation degree of homestead utilization: construction land shows a fixed law, and the proportion of rural homestead shows a gradual decline from high to low [26]. The higher degree of transformation of rural homestead utilization indicates that the potential for remediation of rural idle land is large, and it has a high degree of rural hollowing [9].

Relative diffusion of villages: the relative change of rural population and rural land; if the rate of increase in rural population is slower than that of rural land, this phenomenon indicates that there is a higher degree of rural hollowing, and there is a relatively serious situation of extensive land use [9].

Village population density: this indicator reflects the relative trend of changes in the area of rural settlements and rural population; the larger the population in rural settlement land, the lower the degree of rural hollowing [17].

The proportion of the permanent population: the smaller the value, the more people going out, and the higher the degree of hollowing [17].

Regional economic structure: the evolution trend of hollow villages is closely related to regional population changes, activities, employment and other behaviors; the larger the proportion of non-agricultural industries, the higher the corresponding degree of non-agriculturalization and urbanization of farmers, and the higher degree of rural hollowing [9].

Economic development level: The level of economic development is an important factor leading to rural population migration and land use changes. The higher the level of economic development, the more peasants are separated from agricultural production, the fewer the agricultural employment population, and the higher the degree of rural hollowing [9].

Rural electricity consumption per capita: the higher the per capita electricity consumption in rural areas, the more time people spend at home in rural areas, and the lower the rural hollowing level. 

#### 3.3.2. Calculation of Rural Hollowing Degree

The formula for calculating the degree of rural hollowing is:(1)$$Z=\overset{3}{\underset{i=1}{\sum }}\overset{m}{\underset{i=j}{\sum }}W_{\mathrm{ij}}y_{\mathrm{ij}}$$
where Z is the degree of rural hollowing system; $W_{\mathrm{ij}}$ is the weight of the jth index in the ith type of rural hollowing system; $y_{\mathrm{ij}}$ is the standardized value of the jth index in the ith type of rural hollowing. Before calculating the degree of hollowing in rural areas, this paper first uses the standardization method to standardize the original data and then determine its weight by the entropy method. Finally, a weighted comprehensive standard model is set to calculate evaluation value of rural hollowing degree in Sichuan Province from the perspective of the county. The weights are shown in Table 3.

#### 3.3.3. Spatial Autocorrelation Analysis of Rural Hollowing Degree

From the perspective of geography, things in space are universally connected and exhibit the characteristics of agglomeration. Rural hollowing is also a regional geographical phenomenon with spatial relevance. At present, Moran’s I index and Geary C index are the two primary methods to analyze the degree of spatial autocorrelation, with Moran’s I index being the more commonly used [27]. Therefore, this paper adopts global and local Moran’s I indexes to measure the spatial autocorrelation of rural hollowing degree.

#### 3.3.4. Global Space Autocorrelation

The global spatial autocorrelation describes the degree of rural hollowing at the county level in Sichuan Province from the whole regional space. In this paper, Moran’s I index is adopted, which is defined as follows:(2)$$I=\frac{\sum_{i=1}^{n}\sum_{j=1}^{n}w_{\mathrm{ij}}\left(x_{i}-\bar{x}\right)\left(x_{j}-\bar{x}\right)}{S^{2}\sum_{i=1}^{n}\sum_{i=1}^{n}W_{\mathrm{ij}}},$$

In Equation (2), $S^{2}=\frac{1}{n}\sum_{i=1}^{n}\left(x_{i}-\bar{x}\right)$, $\bar{x}=\frac{1}{n}\sum_{i=1}^{n}x_{i}$, where $x_{i}$ is the observations in i region, n is the total number of regions a, and $w_{\mathrm{ij}}$ is the Spatial weight matrix. The construction principle of $w_{\mathrm{ij}}$ is:(3)$$w_{\mathrm{ij}}=\left\{\begin{array}{ll} 1, & \mathrm{When}\;\mathrm{area}\;i\;\mathrm{and}\;\mathrm{area}\;j\;\mathrm{are}\;\mathrm{adjacent}\\ 0, & \mathrm{When}\;\mathrm{area}\;i\;\mathrm{and}\;\mathrm{area}\;j\;\mathrm{are}\;\mathrm{not}\;\mathrm{adjacent} \end{array}\right.$$

In Equation (3), i = 1, 2, …, n; j = 1, 2, …, m.

Using ArcGIS software, the global Moran’s I index and related indicators of the degree of rural hollowing in Sichuan Province in 2010, 2015 and 2018 were calculated. The global Moran’s I index shows the spatial agglomeration characteristics of rural hollowing degree in Sichuan Province in different periods.

To examine whether there is spatial autocorrelation in n regions by Z, the calculation equation is:(4)$$Z=\frac{\mathrm{Moran}’s\;I-E\left(\mathrm{Moran}’s\;I\right)}{\sqrt{\mathrm{VAR}\left(\mathrm{Moran}’s\;I\right)}}$$

In Equation (4), if Z is significant and positive, it indicates that the spatial distribution of rural hollowing degree at the county level in Sichuan Province is spatially dependent.

#### 3.3.5. Local Spatial Autocorrelation

The local Moran’s I index is used to analyze the local spatial correlation between regions and adjacent regions, and its equation is as follows:(5)$$\mathrm{Moran}’s\;I_{i}=Z_{i}\overset{n}{\underset{j=1}{\sum }}W_{\mathrm{ij}}Z_{j}\;\;\;\left(i\neq j\right)$$

In Equation (5), $Z_{i}=x_{i}-x$, $Z_{j}=x_{j}-x$ is the deviation of the observed value from the mean, $x_{i}$ and $x_{j}$ represent the observed value of the ith unit and the observed value of the jth unit, respectively, $W_{\mathrm{ij}}$ has the same meaning as in Equation (2). If $\mathrm{Moran}’s\;I_{i}$ is positive, it means that the area has a similar spatial clustering to the surrounding area. It has a positive correlation, and the analysis value is proportional to the spatial clustering degree; if it is negative, the opposite is true. It can be divided into four cases: high-value cluster (HH), low-value cluster (LL), low-value–high-value abnormality (LH) and high-value–low-value abnormality (HL).

#### 3.3.6. Correlation Analysis

Empirical research on the influencing factors of rural hollowing mainly focuses on urbanization, rural per capita income, arable land resources, and the circulation and utilization of arable land [8,13,14,28,29]. Combined with previous research and the formation mechanism of rural hollows, this paper selects four indicators of average elevation, per capita arable land, urbanization rate, and industrial output value, to explore the relevant factors of rural hollowing from three aspects: environmental resource endowment, urban-rural gap and urbanization. This paper uses spass24.0 software to do the Pearson correlation test.

## 4. Results

### 4.1. Average Value of Rural Hollowing

It can be seen from Table 4 that the average rural hollowing has a downward trend in general. The average land hollowing decreased by 0.084 from 2010 to 2018, and 148 counties decreased in land hollowing. In contrast, the average population hollowing and economy hollowing increases by 0.014 and 0.040, respectively, during research. There were 168 and 170 counties with an increase in population hollowing and economy hollowing, respectively. Overall, the average comprehensive hollowing degree drops by 0.028; meanwhile, 86 counties had an increase and 92 counties had a decrease in comprehensive hollowing.

### 4.2. Spatial–Temporal Evolution of Rural Hollowing Degree in Sichuan Province

#### 4.2.1. Grading Results of Rural Hollowing Degree

The results of the three subsystems of land, economy, and population jointly determine the degree of rural hollowing. Based on the natural breaks (Jenks) classification method, the three-year grading standards are integrated and unified so that the three-year hollowing degree grades are comparable. Therefore, the degree of rural hollowing in Sichuan Province is divided into five levels: high, higher, moderate, lower and low, as shown in Table 5.

Overall, land and economy and comprehensive hollowing degree have a downward trend, from the perspective of the number of counties, and population hollowing degree is the opposite. The number of counties with a high degree of land hollowing decreased by 44 in 2010, 2015 and 2018, accounting for 24.27%. Counties in higher and moderate degrees of land hollowing increased by 7 and 35, account for 3.94% and 19.66%, respectively. The number of counties in higher population hollowing increased by 28, accounting for 15.73%, and counties with lower and moderate population hollowing decreased by 18 and 11, accounting for 10.11% and 6.18%, respectively, during research. The number of counties with higher economy hollowing drop 17, accounting for 9.55%, and counties with low, lower and high economy hollowing increased by 9, 6 and 5 during research. The number of lower comprehensive hollowing increased by 27, accounting for 15.16% and the number of higher comprehensive hollowing decreased by 27, accounting for 15.16% during research.

#### 4.2.2. Analysis of Time Series Difference of Rural Hollowing Degree

As shown in Figure 3, there are 59 counties with an increase in the degree of hollowing out of rural areas and 119 counties with a decrease from 2010 to 2018. The spatial distribution shows that the degree of hollowing generally increases in the central and southern regions and decreases in the northeast and central and western regions. The areas with an increase in hollowing degree are mainly concentrated in the western Sichuan and Panxi areas. There are 70 counties with an increase in the degree of hollowing out of rural areas and 108 counties with a decrease; from 2010 to 2015, the areas with an increase in hollowing degree are mainly concentrated in northwest Sichuan. There are 86 counties with an increase in the degree of hollowing out of rural areas and 92 counties with a decrease; from 2015 to 2018, the areas with an increase in hollowing degree are mainly concentrated in the central and southwest.

#### 4.2.3. Land Hollowing Degree

It can be seen from Figure 4 that high (0.421–0.513) and higher (0.354–0.421) degrees of land hollowing had an obvious decrease in number from 2010 to 2018, and they were mainly distributed in 2018 in Northwest Sichuan (Higher altitude areas). The number of counties with a high degree of hollowing (0.421–0.513) dropped from 80 in 2010 to 36 in 2018, accounting for 44.94% and 20.22% of the total number of research units, respectively. In terms of spatial distribution, the degree of high hollowing (0.421–0.513) in 2010 and 2015 was widely distributed and scattered, mainly concentrated in the northwest and middle, and gradually decreased in 2015; in 2018, it was mainly distributed in the northwest and south, concentrated in Ganzi Prefecture, Aba Prefecture and Liangshan Prefecture.

The number of moderate hollowing degrees (0.285–0.354) increased from 28 in 2010 to 63 in 2018. In 2010, they were mainly concentrated in the central and western regions of the study area, and in 2018, the range expanded to the east.

The number of low-level hollowing (0–0.001) did not change significantly, with six, eight and six in 2010, 2015 and 2018, accounting for 3.37%, 4.49% and 3.37% of the total number of research units, respectively. The distribution of low-level hollowing degree (0–0.001) is scattered. In 2010, it was relatively concentrated in the northeast, middle and south of the study area; in 2015, it was relatively concentrated in the northeast, middle and southwest of the study area; in 2018, it was relatively concentrated in the study area northeast and south central.

#### 4.2.4. Population Hollowing Degree

Figure 5 shows that the degree of population hollowing gradually increases from west to east, and high (0.111–0.156) and higher (0.092–0.111) degrees of population hollowing have a significant increase in number from 2010 to 2018, mainly concentrated in East Sichuan (areas with better economic development). The degree of population hollowing for each year is mainly concentrated in moderate (0.076–0.092) and higher (0.092–0.111) levels of hollowing. In 2018, the number of higher levels of population hollowing (0.092–0.111) was the largest, accounting for 37.08% of the research unit. In terms of spatial distribution, there was no significant difference in the degree of population hollowing between 2015 and 2018. The low degree of population hollowing (0.039–0.057) was mainly concentrated in the west of Sichuan Province, and Garze Prefecture was the majority. The lower hollowing (0.057–0.076) is distributed in the central part of Sichuan Province, with a north-south zonal distribution, and a few are distributed in the west. Moderate hollowing (0.076–0.092) is mainly distributed in eastern and central and eastern regions. The number of higher hollowing grades (0.092–0.111) is increasing and scattered in the eastern part of the study area. The high hollowing (0.111–0.156) accounts for a small proportion of the research unit scattered in eastern Sichuan.

#### 4.2.5. Economic Hollowing Degree

Figure 6 shows that the degree of economic hollowing has obvious spatial characteristics from west to east, and high (0.179–0.241) and higher (0.145–0.179) degrees of economic hollowing increased from 2010 to 2018, mainly concentrated in Chengdu Plain (Areas with better economic development). The high degree of hollowing (0.179–0.241) gradually increases as the years increase. Additionally, the low degree of hollowing (0.024–0.069) in western China is decreasing. The low degree hollowing (0.024–0.069) was mainly distributed in western Sichuan each year, while a small number of low hollowing (0.024–0.069) was distributed in southern Sichuan in 2018. The distribution of lower hollowing (0.069–0.111) is scattered, mainly in the south and northeast of the study area in 2010; a large number were concentrated in the central region in 2015, with a few in the west. In 2018, it was scattered in the northeast and southeast. The spatial distribution of moderate hollowing (0.111–0.145) showed an unbalanced trend, with scattered distribution in the central and eastern regions each year. The spatial distribution of higher hollowing (0.145–0.179) is mainly concentrated in the central and southern regions. The high hollowing (0.179–0.241) is less scattered in central and southern Sichuan each year.

#### 4.2.6. Comprehensive Rural Hollowing Degree

Figure 7 shows that high (0.674–0.789) and higher (0.603–0.674) degrees of comprehensive hollowing decreased in number from 2010 to 2018, and having a trend of agglomeration in Central Sichuan (Provincial capital and its surroundings).

In 2010, 2015 and 2018, the least of the research area had low hollowing (0.106–0.268), accounting for 3.37%, 3.93% and 2.81% of the research units, respectively. The distribution of low-degree hollowing (0.106–0.268) in each year is scattered, with a little in the middle, south and northeast of 2010. In 2015, a small amount was distributed in the southwest, south, central and northeast of the study area. In 2018, the number of counties with the low hollowing out (0.106–0.268) was the least, with only five research units: Daocheng County, Ganluo County, Enyang District, Jiuzhaigou County and Hanyuan County. In 2010, the number of lower hollowing (0.268–0.512) was small and mainly distributed in the western region. In 2015 and 2018, the level of lower hollowing (0.268–0.512 increased significantly, mainly distributed in the west and northeast. The number of research units with moderate hollowing levels (0.512–0.603) in each year is large, but they are scattered in space and involved in all regions.

The higher hollowing level (0.603–0.674) shows a decreasing trend year by year. In 2010, it was distributed in a belt from northeast to southwest in the eastern region; in 2015, the range of the banded distribution decreased; it was mainly distributed in the central and southern regions of the study area in 2018. The research units with high hollowing (0.674–0.789) showed a trend of decreasing first and then increasing. They were mainly distributed in the central and eastern regions in 2010; decreased and concentrated in the central region in 2015, and increased and mainly concentrated in the central and central and eastern regions in 2018.

### 4.3. Spatial Autocorrelation Analysis of Rural Hollowing Degree

#### 4.3.1. Global Spatial Autocorrelation

As shown in Table 6, Moran’s I index in 2010, 2015 and 2018 were all positive, with a *p*-value of 0.0000, which was significant at the 0.001 level. In general, Moran’s I index shows an increasing trend, indicating that the spatial correlation of rural hollowing degree in Sichuan Province is gradually strengthened. As a regional spatial characteristic and geographical linearity, the formation of rural hollowing is influenced by various factors such as nature, society, economy and policy. With the development of the social economy, the connection between regions is increasingly close, and the homogeneity is gradually strengthened.

#### 4.3.2. Local Spatial Autocorrelation

As shown in Figure 8, the spatial distribution characteristics of clustering degree in each year are obvious. High-value clustering (HH) refers to the areas with high rural hollowing degree and relatively high rural hollowing degree in surrounding areas. In 2010, 2015 and 2018, HH were mainly distributed in Chengdu Plain Economic Zone, and the cluster scope showed a narrowing trend, from 37.08% of the research unit in 2010 to 28.09% in 2018. In 2010, there were a small number of counties with HH in the Southern Sichuan Economic Zone, including Gulin County, Naxi District and Jiangyang District, which became insignificant in 2015 and 2018.

Low-value clustering (LL) refers to the type with a low degree of rural hollowing in itself and a low degree of rural hollowing in its surrounding areas. In 2010, 2015 and 2018, the distribution range of LL was relatively concentrated, accounting for 8.99%, 10.11% and 14.61% of the study area, respectively. In 2010, LL were mainly distributed in Ganzi Prefecture and Panzhihua City. In 2015, it was mainly distributed in Ganzi Prefecture, Aba Prefecture, Dazhou City, Bazhong City and Guangyuan City. In 2018, the LL range of Ganzi Prefecture decreased, the clustering range of northeast Sichuan Economic Zone expanded, and Ruoergai County changed from insignificant to LL.

High-value–low-value abnormality (HL) refers to the area with a high degree of rural hollowing in itself and low degree of rural hollowing in surrounding areas. The distribution range of HL has changed in time, showing an imbalance. In 2010, it was mainly distributed in Huidong, Ningnan, Dechang and Yanyuan counties in the Panxi region, and Malkang city and Jinchuan County in northwest Sichuan. In 2015, the Panxi area showed insignificant characteristics, with HL mainly distributed in Xiangcheng, Daofu, Jinchuan, and Qingchuan counties and Langzhong City. In 2018, it was mainly distributed in Northeast Sichuan Economic Zone, and the Panxi area is Muli Tibetan Autonomous County.

Low-value–high-value abnormality (LH) refers to the area with low rural hollowing degree and high rural hollowing degree in its surrounding areas. In 2010, 2015 and 2018, LH showed an imbalance, accounting for 10.67%, 10.67% and 8.43% of the research units, respectively. In 2010, it was scattered in Chengdu Plain Economic Zone, with a few in the southern Sichuan Economic Zone. In 2015, it was mainly distributed in the periphery of Chengdu Plain Economic Zone and the counties bordering northwest Sichuan ecological demonstration zone and South Sichuan Economic Zone. The number decreased in 2018, mainly distributed in Chengdu Plain Economic Zone and southern Sichuan Economic Zone.

### 4.4. Rural Hollowing Correlation Analysis

As shown in Figure 9 and Table 7, the overall rural hollowing is negatively correlated with the average elevation in Sichuan Province, which is inconsistent with the formation mechanism of rural hollowing. This may be due to poor environmental conditions, poor traffic conditions and poor ability to obtain external information, hindering farmers from leaving the countryside. When examining the regions separately, it is found that the rural hollowing in the Panxi area is positively correlated with the average elevation, which is consistent with the mechanism. The harsh environmental conditions drive farmers to leave the countryside to settle in and work in cities, which in turn leads to the hollowing out of the countryside. This suggests that there may be heterogeneity in the effect of elevation on rural hollowing.

Rural hollowing is negatively correlated with per capita arable land, which is consistent with the formation mechanism of rural hollowing. Less per capita arable land is difficult to satisfy farmers’ desire to get rich, prompting farmers to work in cities, resulting in rural hollowing.

Rural hollowing is positively correlated with urbanization rate and industrial output value, which is consistent with the formation mechanism of rural hollowing. Industrialization and Urbanization provide opportunities for farmers to work and settle in cities, which in turn leads to an increase in rural hollowing.

## 5. Discussion

Carrying out the renovation of hollow villages is an important process to realize rural revitalization, and the calculation and evaluation of the degree of rural hollowing is the basis and premise of the renovation of hollow villages. The results of this research can provide a scientific basis for the renovation of hollow villages and rural revitalization planning in Sichuan Province. We believe that the current high level of rural hollowing is mainly concentrated in the Chengdu Plain and its surrounding areas, so the improvement of rural hollowing should focus on these areas; in the northwestern Sichuan region, we should pay attention to the improvement of land hollowing, and continue to promote rural land improvement work; in the eastern Sichuan area, we should focus on economy hollowing and population hollowing, promote the development of rural industries, and prevent the loss of rural population.

Compared with a previous study [14], the distribution trend of rural hollowing degree in Sichuan Province is consistent, which is high in the east and low in the west. However, the higher degree of rural hollowing in this paper is more concentrated in the Chengdu Plain Economic Zone. In their study, the higher rural hollowing is more concentrated in the eastern Sichuan area. From the theoretical analysis, under the condition of similar resource and environmental endowments, the urbanization and industrialization of the Chengdu Plain Economic Zone should be higher, and the rural hollowing should also be higher. The rural hollowing calculation in this paper should be more accurate. Due to the differences in the multiple indicators involved and the study sites, it is difficult to compare with the results of Zhang Xiupeng et al. [19] to explain the increased accuracy after the improved indicators, but from a theoretical perspective, this research uses rural electricity consumption per capita instead of electricity consumption per household which improved accuracy of the assessment. For example, in 2018, the per capita living electricity consumption in Sichuan Province was 489.37 kilowatts (county-level per capita living electricity consumption data are not available); the per capita electricity consumption in Sichuan Province is 2696.23 kilowatts, which is about 5.5 times that of the per capita living electricity consumption; rural electricity consumption per capita in Sichuan Province is 339.52 kilowatts, which is about 0.7 times that of the per capita living electricity consumption (Source: Sichuan Statistical Yearbook in 2019). Whether it is judged from the value or the proportion, it is more reasonable and accurate to use rural electricity consumption per capita to measure the rural per capita home time.

In terms of relevant factors, rural hollowing is positively correlated with urbanization and negatively correlated with per capita arable land, which is consistent with previous research results [8,13,14]. Urbanization in this paper refers to population urbanization. The higher the degree of urbanization, the more rural residents settle in cities, the greater the loss of rural population, and the more serious the rural population hollowing. In addition, because of the collective property rights system of rural land, the more rural population is lost, the more rural homesteads are abandoned, and the more serious the rural land hollowing. Therefore, the higher the degree of urbanization, the higher the degree of rural hollowing.

The per capita arable land resources are abundant, and the rural residents can get more remuneration from the arable land; the better the development of the primary industry, the lower the degree of the rural economy hollowing. Moreover, rural residents can get more remuneration from arable land, and the lower the willingness of rural residents to go out to work, the less rural population flows, and the lower the degree of rural population hollowing. Therefore, the richer the per capita arable land resources, the lower the degree of rural hollowing.

Rural hollowing is positively correlated with industrialization. The industrialization used in this paper is the industrialized output value. The higher the degree of industrialization, the stronger the pull of industrialization for rural residents, the more likely they are to go out to work, the more serious the loss of rural population, the fewer professional farmers, and the higher the degree of the rural population and economy hollowing. Moreover, after going out to work, the income increases, the expansion capacity of the homestead is enhanced, and the rural land hollowing is more serious. Therefore, the higher the degree of industrialization, the higher the degree of rural hollowing.

On the whole, the altitude is negatively correlated with the rural hollowing, but in the sub-regional test, it is found that the rural hollowing degree is positively correlated with the altitude in the Panxi area. We believe that transportation infrastructure and communication conditions have caused such different results. Panxi is a typical resource-based city with good local transportation and communication conditions, while other high-altitude areas in Sichuan Province have poor transportation infrastructure and communication conditions. When the altitude is higher, the transportation infrastructure and communication conditions are generally worse, the rural areas are more closed, the ability of rural residents to obtain external information and the ability to go out to work are lower, and the loss of rural population is lower; there are more professional farmers, and the lower the degree of the rural population and economy hollowing. Therefore, we believe that when the transportation infrastructure and communication conditions are poor, rural hollowing will be negatively correlated with the altitude; when the transportation infrastructure and communication conditions are good, the opposite is true.

The contribution of this paper is to improve the rural hollowing measurement index and enrich the research on the correlation between altitude and industrialization in rural hollowing. The deficiency of this paper is the evolution of rural hollowing has gone through different stages and changes in time and space, thus requiring a longer evaluation period than the one done by this paper. Due to data availability, this paper only calculated the degree of rural hollowing in 2010, 2015 and 2018. Therefore, continuous time series are needed to evaluate rural hollowing in future studies.

## 6. Conclusions

The temporal and spatial evolution law of rural hollowing degree in Sichuan Province is summarized as follows:(1)From 2010 to 2018, rural hollowing shows a downward trend. The average of comprehensive hollowing drops by 0.028, and the average of land hollowing decreases by 0.084; 59 counties have an increase in the degree of hollowing out of rural areas, and 119 counties have a decrease.(2)During the same study period, the regional characteristics of rural hollowing in Sichuan Province were significant. High and higher degrees of comprehensive hollowing decreased in number from 2010 to 2018, and having a trend of agglomeration in Central Sichuan (Provincial capital and its surroundings). High and higher degrees of land hollowing have an obvious decrease in number from 2010 to 2018, and they mainly distributed in Northwest Sichuan (Higher altitude areas) in 2018. The degree of population hollowing gradually increases from west to east, and high and higher degree of population hollowing have a significant increase in number from 2010 to 2018, mainly concentrated in East Sichuan (Areas with better economic development). The degree of economic hollowing has obvious spatial characteristics from west to east, and high and higher degrees of economic hollowing increased from 2010 to 2018, mainly concentrated in Chengdu Plain (Areas with better economic development).(3)During the same study period, the spatial distribution characteristics of the clustering degree of rural hollowing degree in Sichuan Province were obvious. Among them, the high-value clusters are mainly concentrated in the Chengdu Plain Economic Zone (provincial capital and its surroundings), and the scope of the clusters is shrinking. The low-value clusters are mainly distributed in the Northeast Sichuan Economic Zone and the Northwest Sichuan Economic Zone.(4)Rural hollowing in Sichuan Province is negatively correlated with average altitude and per capita arable land area; rural hollowing is positively correlated with urbanization rate and industrial output value.

## Figures and Tables

**Figure 1 ijerph-19-09075-f001:**
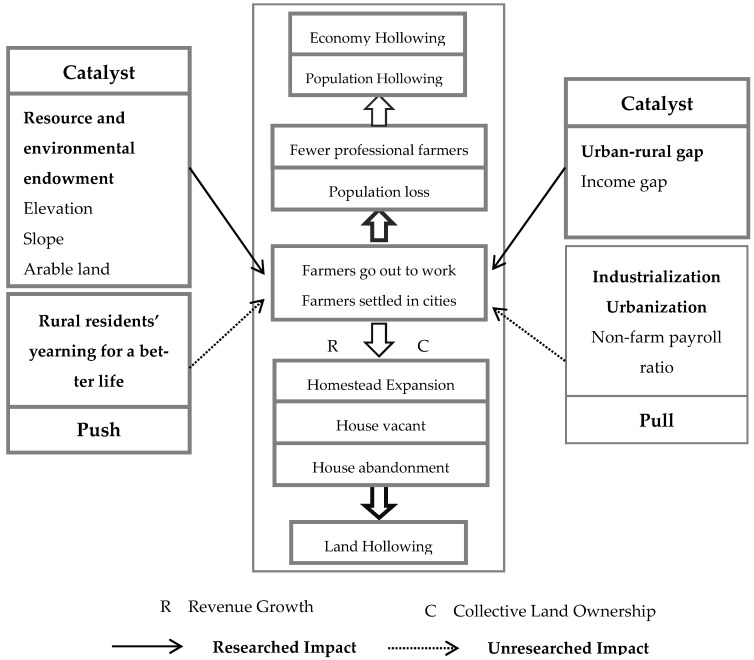
Formation mechanism of rural hollowing.

**Figure 2 ijerph-19-09075-f002:**
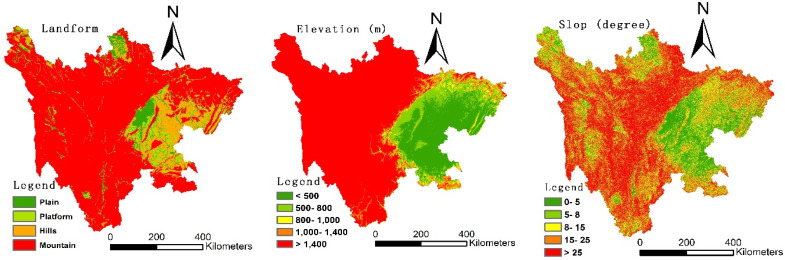
Elevation and slope in Sichuan Province.

**Figure 3 ijerph-19-09075-f003:**
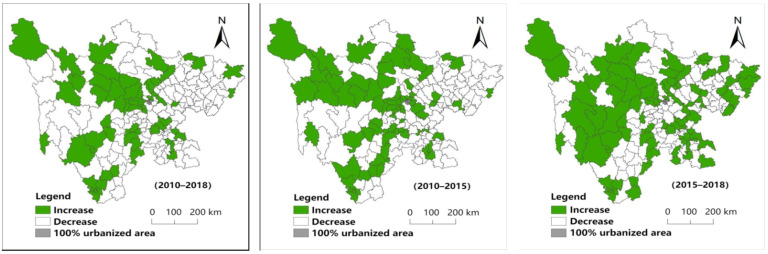
Spatial distribution of changes in the degree of rural hollowing in Sichuan Province.

**Figure 4 ijerph-19-09075-f004:**
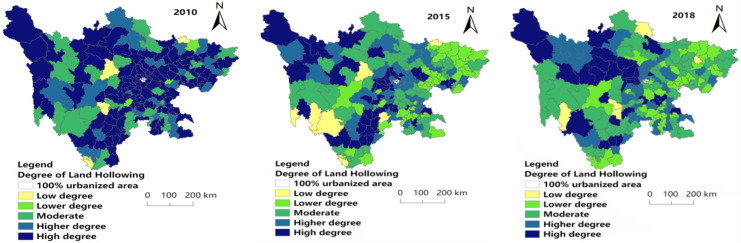
Spatial distribution of land hollowing degree in 2010, 2015 and 2018.

**Figure 5 ijerph-19-09075-f005:**
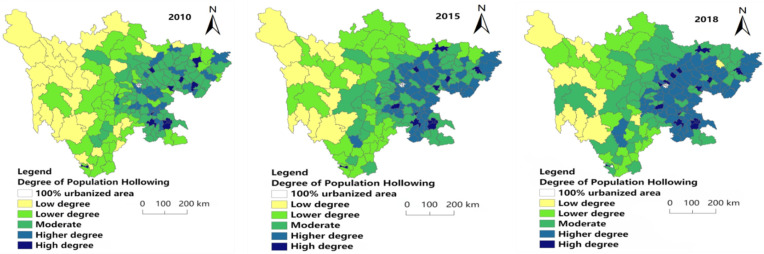
Spatial distribution of population hollowing degree in 2010, 2015 and 2018.

**Figure 6 ijerph-19-09075-f006:**
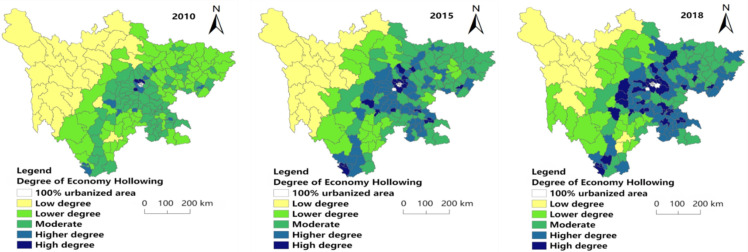
Spatial distribution of economy hollowing degree in 2010, 2015 and 2018.

**Figure 7 ijerph-19-09075-f007:**
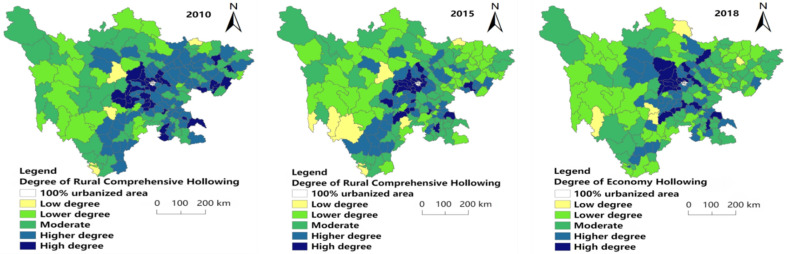
Spatial distribution of comprehensive hollowing degree in 2010, 2015 and 2018.

**Figure 8 ijerph-19-09075-f008:**
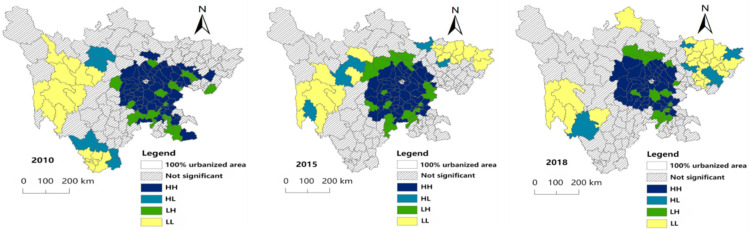
Cluster maps for local spatial autocorrelation of rural hollowing degree in 2010, 2015 and 2018.

**Figure 9 ijerph-19-09075-f009:**
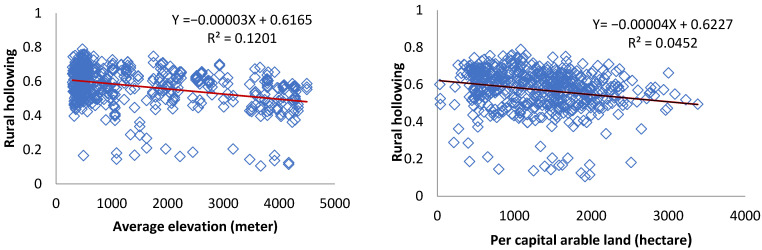
Scatter plot of influencing factors of rural hollowing.

**Table 2 ijerph-19-09075-t002:** Evaluation index system of rural hollowing degree at county scale in Sichuan Province.

Subsystem	Indicators	Measures	Influence	Reference
Land	Transformation degree of homestead utilization	Rural residential land area/construction land area	positive	[9]
	Relative diffusion of villages	(Rand area of rural settlements at the end of the period/Land area of rural settlements at the beginning)/(Rural population at the end of the period/Rural population at the beginning)	positive	[9]
Population	Village population density	Rural population/land area of rural settlements	positive	[17]
	Proportion of permanent population	Rural Resident Population/Household Registered Population	positive	[17]
Economy	Regional economic structure	(secondary and tertiary/GDP	positive	[9]
	economic development level	GDP per capita	positive	[9]
	Rural electricity consumption per capita	Rural electricity consumption/rural population	negative	[19]

Note: The positive index is defined as the larger the value of the index, the higher the degree of hollowing in the countryside, and the negative index is defined as the smaller the value of the index, the higher the degree of hollowing.

**Table 3 ijerph-19-09075-t003:** The weight of the evaluation index of rural hollowing degree in Sichuan Province.

Target	Subsystem	Indicators	Symbol	Weights
Rural hollowing index	Land	Transformation degree of homestead Utilization	L1	0.3114
		relative diffusion of villages	L2	0.2052
	Population	Village population density	P1	0.0255
		Proportion of permanent population	P2	0.1325
	Economy	Regional economic structure	E1	0.1656
		The level of economic development	E2	0.1480
		Rural electricity consumption per capita	E3	0.0118

**Table 4 ijerph-19-09075-t004:** Variation of rural hollowing in 2010, 2015 and 2018.

	Land	Population	Economy	Comprehensive
2010 (average)	0.410	0.079	0.108	0.595
2015 (average)	0.337	0.087	0.138	0.568
2018 (average)	0.326	0.093	0.148	0.567
Variation (10–15)	−0.073	+0.008	+0.030	−0.027
Variation (15–18)	−0.011	+0.006	+0.010	−0.001
Variation (10–18)	−0.084	+0.014	+0.040	−0.028
Increase (10–15)	14	170	173	70
Decrease (10–15)	152	8	5	108
Unchanging (10–15)	11	0	0	0
Increase (15–18)	66	161	153	86
Decrease (15–18)	106	17	25	92
Unchanging (15–18)	5	0	0	0
Increase (10–18)	25	168	170	86
Decrease (10–18)	148	10	7	92
Unchanging (10–18)	5	0	0	0

**Table 5 ijerph-19-09075-t005:** Variation in the amount of rural hollowing at different levels.

	Low		Lower		Moderate		Higher		High	
Land	(0–0.001]		(0.001–0.285]		(0.285–0.354]		(0.354–0.421]		(0.421–0.513]	
2010	6	3.37%	28	15.73%	28	15.73%	36	20.22%	80	44.94%
2015	8	4.49%	39	21.91%	46	25.84%	43	24.16%	42	23.60%
2018	6	3.37%	30	16.85%	63	35.39%	43	24.16%	36	20.22%
Variation	0	0	+2	+1.12%	+35	+19.66%	+7	+3.94%	−44	−24.27%
Population	(0.039–0.057]		(0.057–0.076]		(0.076–0.092]		(0.092–0.111]		(0.111–0.156]	
2010	17	9.55%	47	26.40%	64	35.96%	38	21.35%	12	6.74%
2015	13	7.30%	36	20.22%	57	32.02%	63	35.39%	9	5.06%
2018	15	8.43%	29	16.29%	53	29.78%	66	37.08%	15	8.43%
Variation	−2	−1.12%	−18	−10.11%	−11	−6.18%	+28	+15.73%	+3	+1.69%
Economy	(0.024–0.069]		(0.069–0.111]		(0.111–0.145]		(0.145–0.179]		(0.179–0.241]	
2010	23	12.92%	31	17.42%	53	29.78%	59	33.15%	12	6.74%
2015	17	9.55%	26	14.61%	57	32.02%	59	33.15%	19	10.67%
2018	32	17.98%	37	20.79%	50	28.09%	42	23.60%	17	9.55%
Variation	+9	+5.06%	+6	+3.37%	−3	−1.69%	−17	−9.55%	+5	+2.81%
Comprehensive	(0.106–0.268]		(0.268–0.512]		(0.512–0.603]		(0.603–0.674]		(0.674–0.789]	
2010	6	3.37%	21	11.80%	52	29.21%	69	38.76%	30	16.85%
2015	7	3.93%	48	26.97%	58	32.58%	41	23.03%	24	13.48%
2018	5	2.81%	48	26.97%	55	30.90%	42	23.60%	28	15.73%
Variation	−1	−0.56%	+27	+15.17%	+3	+1.69%	−27	−15.16%	−2	−1.12%

**Table 6 ijerph-19-09075-t006:** Overall Moran’s I index of rural hollowing degree in Sichuan Province.

Time	Overall Moran’s I Index	Z	*p*
2010	0.1459	9.0847	0.0000
2015	0.2156	13.2355	0.0000
2018	0.2189	13.4284	0.0000

**Table 7 ijerph-19-09075-t007:** Correlation of influencing factors of rural hollowing in Sichuan Province.

	Rural Hollowing Degree		
	Pearson Correlation	Sig. (2-Tailed)	N
Average elevation	−0.213 **	0.000	534
Per capita arable land	−0.347 **	0.000	534
Industrial output	0.258 **	0.000	534
Urbanization rate	0.266 **	0.000	534

** Correlation is significant at the 0.0.1 level (2-tailed).

## Data Availability

The datasets used and/or analyzed during the current study are available from the corresponding authors upon a reasonable request.

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
