# Peer review of "Spatial–Temporal Features and Correlation Studies of County Rural Hollowing in Sichuan"

_ijerph, 2022, doi:10.3390/ijerph19159075_

Round 1
Reviewer 1 Report
An excellent paper and well worth publishing

Author Response
Thanks for your review.
Reviewer 2 Report
Rural hollowing is a phenomenon worthy of study in the process of rural development. This paper evaluates the spatio-temporal characteristic of County Rural hollowing in Sichuan Province, and makes a correlation analysis. On the whole, the paper is useful and valuable, but there are still many problems.
1. The literature review needs to be strengthened. The current literature review only lists the research status of evaluation indicators of hollow villages with simple comments, and the theoretical value of the paper is not prominent. Compared with previous research, what is the improvement and marginal contribution of this paper?This needs to be clarified and highlighted.
2. The paper tends to be descriptive and it might be improved by a more critical tone. The paper is relatively short and succinct. The analysis might be easily expanded.
3. The quality of the figures should be improved.
4. The discussion section should be further enriched and deepened. The paper has only briefly discussed the mechanism of rural hollowing. What mechanism will affect the hollowing out of rural areas and how to further excavate it? These are the parts worthy of in-depth discussion. I suggest that theoretical mechanisms can be further analyzed in the discussion part, which can improve the value of the paper.
5. The correspondence between policy implications and analysis results is not clear enough. Some targeted policy implications should be proposed and fit into the needs of local rural community.
Author Response
please see the attachment.

This manuscript is a resubmission of an earlier submission. The following is a list of the peer review reports and author responses from that submission.
Round 1
Reviewer 1 Report
The hollowing out of rural areas is a problem worthy of study in the process of rural development. Taking Sichuan Province as a case, the author attempts to evaluate the spatio-temporal change process of County Rural hollowing in Sichuan Province. On the whole, the idea of the article is relatively clear, but there are still many problems.
- The theoretical value of this paper needs to be improved. For the selection of case area, we should not simply say that there is no research here, but pay more attention to the value of carrying out research in this area. The literature review only lists the current evaluation indicators of hollow villages, without relevant comments, and the theoretical value of the paper is not prominent. Here, please allow me to ask you a very simple question. Why do you have to study and adopt similar indicators after so many people have carried out relevant evaluation index research? Is their indicator selection not scientific enough? Or does this paper adopt more advanced evaluation methods? At present, we can't see the theoretical value of this research.
- The innovation of this paper mentioned in the discussion part cannot be recognized by us. On the one hand, the scientific of the establishment of the index system needs to be strengthened. Since this paper takes the index system as the innovation point, the scientific and rationality of the evaluation system is the key of this paper. It can't be summarized in one sentence, such as “taking the actual situation of Sichuan Province and the availability of data into account”. We can't just list a few references and say that relevant indicators are selected according to the actual situation. The selection basis of each index needs to be explained in detail. Why only these indicators are selected? Can we fully reflect the situation of hollow villages in Sichuan Province? All need to be demonstrated. Instead of just listing a few references, it is said to be selected according to the actual situation. On the other hand, at present, the index system of this paper is selected by referring to the existing relevant literature, and we don't see any innovation.
- The concept of economic hollowing and related indicators need to be discussed. What does economic hollowing mean? Judging from your evaluation indicators, they are just conventional indicators such as economic structure, economic development stage and income. That is, the conventional economic development indicators, why use these to represent economic hollowing out?
- In the discussion part, the temporal and spatial differentiation of hollowing out is attributed to natural and economic conditions, but the relevant evaluation of economic hollowing has been established in the index system you have established, and the evaluation indicators are all conventional economic development indicators. Here, the economic conditions are used to explain the reason, isn't it a kind of logical nesting?
- In Figure 1, there is a repetition of two pictures, and there are two pictures from 2010 to 2015.
Reviewer 2 Report
Evolution of Spatial and temporal pattern of rural hollowing degree in Sichuan Province.
Summary and overview
This is an important paper as it assesses the use of rural development in terms of its sustainability. The first observation might be to invite a simpler title and a clearer explanation of what the paper covers.
The paper is focused on China’s urbanisation and sustainable goals , the case study is focused on Sichuan Province. Perhaps it would be useful to include an explanation of why Sichuan Province is chosen as the main focus of the study. It would be useful to have some clarity on the meaning of “rural hollowing” that would help a less specialised audience.
On page 2 there is a useful table of the main indicators that apply and the areas where there is rural hollowing. It might be useful to include some photographs or at least some maps showing the areas where rural hollowing might occur. One interesting point is that some scholars use the average us of power as an evaluative tool to decide how to measure rural hollowing and its extent. In fact the paper has as its main aim to consider the methodology, the different measurements and systems that might help to evaluate and assess what rural hollowing means. The paper evaluates 178 counties in Sichuan Province. The period extended from 2010 to 2018. Its main findings from the research show the many variables at work and how these may be tabulated and assessed.
The paper does not spell out precisely the value of the exercise and why it is needed. Some attention is needed to the question of whether the research informs policy makers as to what to take into account. This is especially important in terms of agriculture and natural resources. There needs to be a much more critical analysis of the benefits and detriments of the findings. Understanding the “degree of hollowing” needs to fit into the policy process. This is unclear. We also need to know to what extent financial support might be available linked to government funding, and the differences between public bodies and private commercial organisations.
Rural policy making is essential we need to understand how these fit into the role of central government and local community needs.
Organisation of the Paper
The paper and methodology are logical and well – organised. The paper tends to be descriptive and it might be improved by a more critical tone. The paper is relatively short and succinct. The analysis might be easily expanded.
Conclusions
The paper is useful and valuable. It needs some tighter and deeper analysis of the policy implications of the research. This would be achieved very easily with a more focused set of conclusions and a sharper focused introduction.
Reviewer 3 Report
Dear authors, the quality of the figures should be improved.
Round 2
Reviewer 1 Report
The author answers the relevant questions and makes corresponding modifications in the paper, which improves the overall scientificity of this paper. We generally agree with the explanations and modifications made by the author, but there are still some problems worth discussing:
1. We suggest that the content of the discussion should also be taken as one of the conclusions and reflected in the abstract.
2. Since the innovation proposed in the paper is "this research uses rural electricity consumption per capita instead of electricity consumption per household which improved accuracy of assessment." We suggest that in the discussion, it is best to explain how the current research is different from the previous research after changing this indicator.
